# Polymeric Delivery Systems as a Potential Vaccine against Visceral Leishmaniasis: Formulation Development and Immunogenicity

**DOI:** 10.3390/vaccines11081309

**Published:** 2023-07-31

**Authors:** João Guilherme Lino da Silva, Ana Alice Maia Gonçalves, Liliam Teixeira Oliveira, Giani Martins Garcia, Maurício Azevedo Batista, Ludmila Zanandreis de Mendonça, Kelvinson Fernandes Viana, Rita de Cássia Oliveira Sant’Ana, Otoni Alves de Oliveira Melo Júnior, Denise Silveira-Lemos, Walderez Ornelas Dutra, Olindo Assis Martins-Filho, Alexsandro Sobreira Galdino, Sandra Aparecida Lima de Moura, Vanessa Carla Furtado Mosqueira, Rodolfo Cordeiro Giunchetti

**Affiliations:** 1Laboratory of Biology of Cell Interactions, Department of Morphology, Federal University of Minas Gerais (UFMG), Belo Horizonte 31270-901, Brazil; oniloaoj@aluno.ufsj.edu.br (J.G.L.d.S.); anafish@hotmail.com (A.A.M.G.); mauricio.batista@sebsa.com.br (M.A.B.); ludmilazm@ufmg.br (L.Z.d.M.); kelvinson.viana@unila.edu.br (K.F.V.); rita.santana@ufv.br (R.d.C.O.S.); otoni@biof.ufrj.br (O.A.d.O.M.J.); waldutra@icb.ufmg.br (W.O.D.); 2Nucleus for Research in Biological Sciences (NUPEB), Federal University of Ouro Preto, Ouro Preto 35400-000, Brazil; 3Laboratory of Pharmaceutics and Nanotechnology (LDGNano), School of Pharmacy, Federal University of Ouro Preto, Ouro Preto 35400-000, Brazil; liliamto@gmail.com (L.T.O.); giani.garcia@univaco.edu.br (G.M.G.); mosqueira@ufop.edu.br (V.C.F.M.); 4Integrated Research Group on Biomarkers, René Rachou Research Center, Oswaldo Cruz Foundation, Belo Horizonte 30190-009, Brazil; denise.lemos@gmail.com; 5National Institute of Science and Technology in Tropical Diseases, INCT-DT, Salvador 40110-060, Brazil; 6Laboratory of Diagnosis and Monitoring Biomarkers, René Rachou Research Center, Oswaldo Cruz Foundation, Belo Horizonte 30190-009, Brazil; oamfilho@cpqrr.fiocruz.br; 7Laboratory of Microorganism Biotechnology, Federal University of São João Del-Rei (UFSJ), Midwest Campus, Divinópolis 35501-296, Brazil; asgaldino@ufsj.edu.br; 8Laboratory of Biomaterials and Experimental Pathology, Institute of Exact and Biological Sciences, Federal University of Ouro Preto, Ouro Preto 35402-136, Brazil; sandramoura@ufop.edu.br

**Keywords:** visceral leishmaniasis, submicrometric particles, vaccine, polymeric nanoparticles

## Abstract

Recent studies suggest that the association of antigens in microparticles increases the anti-*Leishmania* vaccine immunogenicity. This study aims to investigate the in situ effect of the adjuvant performance consisting of chitosan-coated poly(*D*,*L*-lactic) acid submicrometric particles (SMP) and analyze the inflammatory profile and toxicity. Two formulations were selected, SMP^1^, containing poly(*D*,*L*-lactide) (PLA) 1% *wt*/*v* and chitosan 1% *wt*/*v*; and SMP^2^, containing PLA 5% *wt*/*v* and chitosan 5% *wt*/*v*. After a single dose of the unloaded SMP^1^ or SMP^2^ in mice, the SMPs promoted cell recruitment without tissue damage. In addition, besides the myeloperoxidase (MPO) activity having demonstrated similar results among the analyzed groups, a progressive reduction in the levels of N-acetyl-*β-_D_*-glucosaminidase (NAG) until 72 h was observed for SMPs. While IL-6 levels were similar among all the analyzed groups along the kinetics, only the SMPs groups had detectable levels of TNF-α. Additionally, the *Leishmania braziliensis* antigen was encapsulated in SMPs (SMP^1^Ag and SMP^2^Ag), and mice were vaccinated with three doses. The immunogenicity analysis by flow cytometry demonstrated a reduction in NK (CD3^−^CD49^+^) cells in all the SMPs groups, in addition to impairment in the T cells subsets (CD3^+^CD4^+^) and CD3^+^CD8^+^) and B cells (CD19^+^) of the SMP^2^ group. The resulting data demonstrate that the chitosan-coated SMP formulations stimulate the early events of an innate immune response, suggesting their ability to increase the immunogenicity of co-administered *Leishmania* antigens.

## 1. Introduction

Leishmaniasis comprises a group of endemic neglected diseases caused by flagellated protozoa of the genus *Leishmania* [1]. Visceral leishmaniasis (VL), the most severe form, is caused by *L. donovani* and *L. infantum* (synonym *Leishmania chagasi* in South America), and the parasites are transmitted to vertebrate hosts during the blood meal of female sandflies of the genera *Phlebotomus* or *Lutzomyia* in the Old and New World, respectively [2,3]. VL has an annual incidence of approximately 50,000–90,000 new cases [4]. Of these, more than 90% are concentrated in ten countries, among them Brazil [4], which corresponds to approximately 97% of cases in the Americas [5].

The domestic dog is considered the main reservoir, mainly due to the high parasite load found in the skin of this animal [6] and to the fact that canine visceral leishmaniasis (CVL) usually precedes human cases [7]. Thus, dog vaccination is the only viable way to control zoonotic diseases [8]. In response to such need, and as a result of the researchers’ efforts, there are currently the following three commercial anti-CVL vaccines: CaniLeish^®^ and LetiFend^®^, available in Europe, and Leish-Tec^®^, marketed in Brazil. However, the efficacy of these vaccines is moderate, and, therefore, there is still a pressing need to develop a vaccine with better efficacy to add to the arsenal of health control programs against the disease.

The potency of a vaccine candidate can be enhanced by using adjuvants that polarize the immune response to Th1 or Th2 [9]. Among these, chitosan is a linear cationic polysaccharide composed of randomly distributed β-(1→4)-linked *D*-glucosamine (deacetylated unit) and N-acetyl-*D*-glucosamine (acetylated unit). It is a semisynthetic polymer resulting from the deacetylation of natural chitin extracted from shells of crustaceans, which has shown promising results in the preparation of submicronic particulates designed for antigen delivery. Encapsulation employing chitosan particles are generally nano and microstructured and the antigen may be anchored to their surface or entrapped into their polymeric matrix [10,11]. Different studies have shown that chitosan-based formulations have a large capacity for bioadsorption, are considered biocompatible, present low toxicity by different administration routes and, due to simple production techniques, present low cost. Moreover, chitosan-based formulations have the ability to induce a strong immune response [10,12,13,14,15,16]. In fact, a recent study showed that a chitosan microparticle based-vaccine was able to induce lymphoproliferation and reduce the parasite load in *L*. *infantum*-infected and vaccinated animals [17].

In addition, polymer-based nanoparticles are competent in boosting the quality and magnitude of immune responses [18]. Poly-*D*,*L*-lactic acid (PLA) is a biodegradable polymer and has been used as a platform for developing new vaccine candidates for various infections [11,16,19,20,21,22]. Studies have demonstrated the potential effects of PLA-based systems for antigen encapsulation, showing maintenance antigens activity as well as enhanced and sustained immune response after a single-dose vaccine [23,24]. The use of nanoparticles in VL has shown promising results, with the activation of a protective cellular immune response and reduction in parasite load [25,26,27].

The authors have developed an LBSap vaccine formulation composed of *L. braziliensis* promastigote proteins plus saponin as an adjuvant, which proved harmless and safe for administration in murine, hamster and canine models against *L. infantum* infection. Furthermore, this vaccine was able to induce an increase in total IgG, IgG1 and IgG2 anti-*Leishmania* levels as well as CD5^+^, CD4^+^ and CD8^+^ T lymphocytes (TL) levels and CD21^+^ B lymphocytes in dogs [28,29]. Analyses of immunoprotection using *L. braziliensis* as a heterologous vaccine and *L. infantum* experimental challenge demonstrated a lymphocyte activation profile related to the resistance phenotype, with increased CD5^+^ and CD8^+^ as well as CD14^+^ monocytes, leading to a great lymphoproliferative *Leishmania*-specific activity and a reduced parasite load [30,31]. Therefore, the LBSap has been considered a promising vaccine candidate against CVL. As such, the aim of this study is to report the development of new chitosan-coated PLA polymeric submicrometric particles (SMP) and evaluate their capacity to stimulate early immune response events. In addition, the safety and toxicity of different SMP are analyzed, along with kinetic analysis of the inflammatory profile in the dermis of mice. Subsequently, the effects of *Leishmania braziliensis* antigens associated with SMP are systemically evaluated as a potential VL vaccine.

## 2. Materials and Methods

### 2.1. Materials

PLA, poly(*D*,*L*-lactide), Mw 75,000–120,000 g/mol (inherent viscosity of 0.55–0.75), and low molecular weight chitosan (75–85% of deacetylation) were purchased from Sigma-Aldrich (Sigma-Aldrich Co., St. Louis, MO, USA). Antigens that compose the LBSap vaccine, used as a control of immunogenicity, were obtained as previously described [28]. Acetone, acetic acid and trichloromethane (analytical grade) were provided by Tedia (Rio de Janeiro, Brazil), and other chemicals were commercially available reagent-grade and used without further purification. Milli-Q water was purified using a Simplicity^®^ System (Millipore, Bedford, MA, USA), which was used throughout the experiments.

### 2.2. Methods

#### 2.2.1. Development and Characterization of Submicrometric Particles (SMP)

The polymeric colloidal suspensions were prepared by interfacial polymer deposition following solvent displacement, as previously described [32], with modifications [33]. Briefly, different concentrations of PLA were tested (concentration ranges from 0.5% to 5% *wt*/*v*) and the polymer was completely dissolved in 6 mL of acetone. This organic solution was poured into 12 mL of external aqueous phase containing chitosan solution (concentration ranges from 0.5% to 5% *wt*/*v*), previously solubilized in acetic acid 0.05 M. The final mixture was maintained under magnetic stirring for 10 min. The solvent and part of the water were evaporated under reduced pressure to a final volume of 3 mL.

Antigen-containing nanoparticles were prepared similarly to that described above, adding 14 mg of crude antigen (Ag) in the aqueous phase. The antigen sample was previously sonicated using a titanium probe (40% potency) in an ice bath before being added to the aqueous phase. Two types of particles were selected, the small submicrometric nanoparticles (SMP^1^) and bigger ones (SMP^2^). The physicochemical characterization was performed only in freshly prepared samples.

Dynamic light scattering analyses were performed at 90° angle to determine the mean average size (hydrodynamic diameter) and polydispersion of the particle population in three different nanoparticle batches (Nanosizer N5 Plus, Beckman Coulter Inc., Irving, TX, USA). The zeta potential measurements (ζ) were determined by Doppler laser electrophoresis (Zetasizer 3000 HS, Malvern Instruments, Malvern, UK) after 1:1000 dilution in 1 mM NaCl, maintaining the conductivity close to 120 ± 20 mS/cm. The values reported are the mean ± standard deviation of at least three different batches of each nanoparticle formulation. 

After preparation, the SMP formulations were aliquoted in small cryotubes, stored under −80 °C and freeze-dried without cryoprotectants in freeze-drying equipment (Liobras, model 101, São Carlos, Brazil). Freeze-dried samples were stored at −20 °C until use. To test the formulations, all of them were resuspended in saline at an appropriate dilution to obtain 60 µg of antigen/100 µL (antigen-loaded SMP), or to an equivalent mass of antigens (blank-SMP). These samples were sonicated to facilitate dispersion in an automatic ultrasonic water bath (Quimis, model Q335D2, Campanário, Diadema, SP, Brazil) and used throughout the experiments in biological tests.

#### 2.2.2. Determination of Antigen Encapsulation Efficiency in SMP

The encapsulation efficiency (EE%), the percentage of antigen encapsulated during the process, was determined in freshly prepared liquid samples. Antigen association with SMP was calculated by the difference between the total antigen in the final particle dispersion and the free antigen present in the external aqueous phase, divided by the initial antigen weight used to prepare SMP (Equation (1)), as previously reported [34,35]. The antigen payload is the weight of the antigen associated with SMP divided by the total weight of SMP, calculated following Equation (2).

The antigen was quantified using the Bicinchoninic Acid method for protein dosage (BCA Protein Assay Kit Pierce^®^, ThermoFisher Scientific, Waltham, MA, USA), according to the manufacturer’s instructions using a spectrophotometer (Helios α, Thermo Corporation, USA) at 562 nm.

The antigen not associated with SMP and presented in the external aqueous phase of particles was assessed by ultracentrifuging 3 mL of the SMP formulation at 20,000× *g* for 5 min. The supernatant was collected and the pellet (SMP) was resuspended again with 3 mL of water. This procedure was repeated twice to wash all antigens not adsorbed in SMP. Then the supernatants were pooled and analyzed by the BCA method as described above.

The total antigen associated with the SMP was assessed by dissolving the SMP formulation in trichloromethane/water (1:1) and mixing it in a vortex for 5 min to disrupt the SMP. Then, this solution was centrifuged and the aqueous phase (supernatant) was separated. The antigen released from SMP was then quantified using the BCA method. A similar procedure was conducted with blank-SMP particles and produced no absorption at 562 nm with the BCA method. The weight of each component of SMP formulation is described in Table 1.
(1)$$\mathrm{Encapsulation}\;\mathrm{efficiency}\left(\%\right)=\frac{{\mathrm{Antigen}\;\mathrm{weight}}^{\mathrm{final}\;\mathrm{product}}-{\;\mathrm{Antigen}\;\mathrm{weight}}^{\mathrm{external}\;\mathrm{aqueous}\;\mathrm{phase}}\;}{\mathrm{Antigen}\;\mathrm{weight}\;\mathrm{fed}\;\mathrm{in}\;\mathrm{the}\;\mathrm{formulation}}\times 100$$
(2)$$\mathrm{Payload}\;\left(\%\right)\;wt/wt=\frac{\mathrm{EE}\%\;\times \;\mathrm{Antigen}\;\mathrm{weight}\;\mathrm{fed}\;\mathrm{in}\;\mathrm{the}\;\mathrm{formulation}}{\mathrm{Total}\;\mathrm{weight}\;\mathrm{of}\;\mathrm{particles}\;\left(\mathrm{polymer}\;+\;\mathrm{antigen}\right)}$$

#### 2.2.3. Determination of Inflammatory Profile of Blank-Submicrometric Particles (SMP) in Mice Dermis

##### Animals and Immunization Protocol

Female BALB/c mice, 6–8 weeks old, were purchased from the Rene Rachou Institute’s Production Vivarium–Minas Gerais state and were kept in ventilated racks with food and water ad libitum throughout the study. The protocol for the animal experiments was approved by the Ethics Committee on Animal Use (Protocol 2011/13) at the Federal University of Ouro Preto.

Mice were inoculated with an intradermal injection on the dorsal region with a single dose of submicrometric particles and evaluated at several time points (1, 12, 24, 48 and 72 h) in two independent experiments. A visibly raised cutaneous swelling was regarded as evidence of successful intradermal administration. The intradermal route was used to provide a compartmentalized response, and it was assessable for the in situ evaluation. To evaluate the effects caused by the response to the submicrometric particles, animals were divided into the following three experimental groups (*n* = 5 animals/group/time): control (CT) group, inoculated with 100 µL/dose of saline; SMP^1^ group, inoculated with 100 µL/dose of SMP^1^ formulation; and SMP^2^ group, inoculated with 100 µL/dose of SMP^2^ formulation. Animals were submitted to anesthesia using ketamine (200 mg/kg) and xylazine (16 mg/kg), for the removal of skin samples of the size of 0.5 cm. The animals were euthanized by cervical dislocation for histological analysis, cytokine and soluble proteins assessment. The clinical status of the animals was analyzed throughout the experiment, including behavioral aspects of the animals (pain or irritability) in addition to in situ changes.

##### Histological Examination

The histological examination was conducted as previously described [36]. Skin biopsies from the inoculation sites were fixed in 10% formalin, processed, embedded in paraffin, cut by microtome into 5 μm sections and fixed on glass slides. The samples on the glass slides were stained with hematoxylin and eosin for the quantification of cellular infiltration, photographed and analyzed by using image analysis by morphometry. Inflammatory cells present in the skin were counted through the acquisition of random images, totaling 20 images, using a Leica DM5000B microscope with Leica Application Suite (version 2.4.0 R1, Leica Microsystems Ltd., Heerbrugg, Switzerland). The image analyses were realized through counting all cell nuclei employing Leica Qwin V3 (Leica Microsystems Ltd., Heerbrugg, Switzerland).

##### Cytometric Bead Array

IL-6 and TNF-α cytokine levels were measured by cytometric bead array (Becton Dickinson, Moutain View, CA, USA), according to the manufacturer’s recommendations. In summary, the skin sample was homogenized using a tissue homogenizer (Homo mix) in 1 mL of a specific buffer for cytokine extraction (0.4 M NaCl, 0.05% Tween 20, 0.5% bovine serum albumin, 0.1 mM phenyl methyl sulfonyl hydrofluoric, 0.1 mM benzethonium chloride, 10 mM ethylenediamine tetra-acetic acid [EDTA] and 20 KI of aprotinin). The skin homogenates were centrifuged at 10,000× *g* for 10 min at 4 °C and supernatants were stored at −70 °C before analysis. The cytokines’ standard curves were plotted and the concentrations of each sample were calculated using the FCAP software array v.1.0.2 (Becton Dickinson, Moutain View, CA, USA).

##### Determination of Myeloperoxidase and N-Acetyl-_D_-Glucosaminidase Activities

The extent of neutrophil accumulation was measured by assaying myeloperoxidase (MPO) activity as previously described [36]. Briefly, after processing the skin, as described above, a part of the corresponding pellet was weighed, homogenized in 2 mL of buffer (0.1 M NaCl, 0.02 M Na_3_PO_4_, 0.015 M Na_2_-EDTA, pH 4.7), and centrifuged at 12,000× *g* for 10 minutes at 4 °C. The pellets were resuspended in 0.05 M sodium phosphate buffer, pH 5.4, containing 0.5% hexa-1,6-bis-decyltrimethylammonium bromide (HTAB). The reaction was stopped by the addition of 50 µL of H_2_SO_4_ (4 M). The MPO activity was assayed by measuring the change in absorbance (optical density, OD) at 450 nm using 3,3′-5,5′-tetramethylbenzidine (TMB). Results were expressed as the change in OD per mg of tissue.

The infiltrated mononuclear cells were quantified by measuring the levels of the lysosomal enzyme N-acetyl-*β-_D_*-glucosaminidase (NAG), present in activated macrophages [36]. Briefly, after processing the skin, as described above, part of the pellet was kept for this assay. These pellets were weighed, homogenized in NaCl solution (0.9% *w*/*v*) containing 0.1% *v*/*v* Triton X-100 (Promega) and centrifuged at 3000× *g* for 10 min at 4 °C. Samples of the resulting supernatant (100 µL) were incubated for 10 min with 100 µL p-nitrophenyl-N-acetyl-*β-_D_*-glucosaminide (Sigma-Aldrich, St. Louis, MO, USA). The reaction was stopped by the addition of 100 µL of 0.2 M glycine buffer, pH 10.6. Hydrolysis of the substrate was determined by measuring the absorption at 400 nm. NAG activity was expressed as the change in OD per milligram of wet tissue.

#### 2.2.4. Systemic Evaluation of *Leishmania braziliensis* Antigen Encapsulated SMPs

##### Immunization Protocol

To evaluate the *L. braziliensis* antigens formulated in SMPs, BALB/c mice were vaccinated with three subcutaneous doses at 14-day intervals. The subcutaneous route was used as a common compartment for immunization to trigger a systemic immune response. The antigen doses were standardized with 60 µg of antigen for all particle formulations. The animals were divided into the following six experimental groups (*n* = 5 animals/group): CT group, inoculated with 100 µL/dose of saline; SMP^1^, inoculated with 100 µL/dose of SMP^1^ (124 µg of SMP^1^); SMP^2^, inoculated with 100 µL/dose of SMP^2^ (460 µg of SMP^2^); SMP^1^Ag, inoculated with 100 µL/dose of *L. braziliensis* antigen-encapsulated in SMP^1^ (60 µg of *L. braziliensis* antigen loaded in 184 µg of SMP^1^); SMP^2^Ag, inoculated with 100 µL/dose of *L. braziliensis* antigen-encapsulated SMP^2^ (60 µg of *L. braziliensis* antigen loaded in 520 µg of SMP^2^); and *L. braziliensis* antigen plus saponin as adjuvant (LBSap formulation, prepared as [31]), inoculated with 100 µL/dose of LBSap vaccine (60 µg of *L. braziliensis* antigen plus 50 µg saponin adjuvant). 

##### Immunophenotyping of Blood Cells by Flow Cytometry

The blood cell immunophenotyping was performed by flow cytometry 14 days after the third dose. The monoclonal antibodies anti-CD14 (FITC Rat anti-Mouse CD14, clone Sa2-8, eBioscience), anti-CD3 (PE Hamster anti-Mouse CD3, clone 145-2C11, Biolegend), anti-CD4 (PercP-Cy^TM^ 5.5 Rat anti-Mouse CD4, clone RM4-5, BD Pharmingen^TM^), anti-CD8 (FITC Rat anti-MouseCD8a, clone 5H10, Catalg), anti-CD19 (FITC Rat anti-MouseCD19, clone 6D5, Catalg) and anti-CD49b (FITC anti-Mouse CD49b, clone HMA2/E00340229, e-Bioscience) were used. Briefly, the antibodies were added to polystyrene tubes containing 25 µL of peripheral whole blood collected in EDTA. After homogenization in vortex, the suspensions were incubated for 30 min at room temperature under light shelter. After erythrocytes lysis, the samples were centrifuged at 400× *g* for 7 min at room temperature. The supernatant was discarded, and the leucocytes washed with phosphate-buffered saline, pH 7.4, and centrifuged at 400× *g* for 7 min at room temperature. Posteriorly, the leukocytes were fixed with 200 µL of FACS FIX solution (10.0 g/L paraformaldehyde, 10.2 g/L sodium cacodylate and 6.65 g/L sodium chloride, pH 7.2) and stored at 4 °C prior to flow cytometric acquisition. Flow cytometric measurements were performed on FACSCalibur^®^ (Becton Dickinson). A total of 15,000 events were acquired in CELLQuest^®^ (Franklin Lakes, NJ, USA), and the Flow Jo Software (Flow Cytometry Analysis Software 7.6., Tree Star, Inc., Ashland, OR, USA) was used for data analyses (Appendix A). Nonspecific binding was monitored with fluorochrome-labeled isotypes to provide valid negative controls. Autofluorescence was monitored using a negative control in which cell suspension was incubated with dilution and wash buffer in the absence of fluorochrome-labeled antibodies.

### 2.3. Statistical Analyses

All statistical analyses were performed using GraphPad Prism 5.0 software (Prism Software, CA, USA). Physicochemical data of SMP characterization were analyzed by unpaired *t*-test. Data normality was demonstrated by the Kolmogorov–Smirnoff test. The analysis of variance (ANOVA) tests, followed by the Tukey multiple comparisons test, were used to compare between the groups. Statistical differences were considered significant when the *p*-value was less than 0.05. 

## 3. Results

### 3.1. Physicochemical Characterization of Particles

Biodegradable polymeric SMPs were developed and prepared by the nanoprecipitation method [36]. The weight of PLA and chitosan used in formulations varied in order to produce submicrometric particles with different sizes and capable of entrapping the vaccinal (*L. braziliensis*) antigen (Table 1). The blank (without antigens)-SMP particles containing 1% *wt*/*v* of PLA and coated with 1% *wt*/*v* of chitosan, designated SMP^1^, presented a lower size (347.9 nm and +71.6 mV of zeta potential). The blank-SMP, which contains 5% *wt*/*v* of PLA and 5% *wt*/*v* of chitosan, displayed a higher size (802 nm and +68.4 mV of zeta potential) and it was designated SMP^2^ (Table 1 and Appendix A). The positively charged surface of particles confirms the chitosan location at the surface of the particles, as schematically represented in Figure 1.

**Table 1 vaccines-11-01309-t001:** Physicochemical characterization of the *L. braziliensis* antigen formulations.

Formulations (1 mL)	PLA (mg)	Chitosan (mg)	Crude Antigen (mg)	Hydrodynamic Diameter (nm)	PdI	Zeta Potential (mV ± SD)	Encapsulation Efficiency (%)	Payload (%) *wt*/*wt*
Ag	(-)	(-)	14	423.9 ± 39.8	0.354	−35.7 ± 0.6	(-)	(-)
Blank-SMP^1^	10	10	0	347.9 ± 53.7	0.556	+71.6 ± 2.2	(-)	(-)
SMP^1^Ag	10	10	14	416.2 ± 49.5	0.521	+57.0 ± 1.5 **	69.15 ± 31.60	28.47
Blank-SMP^2^	50	50	0	802.0 ± 29.7	0.263	+68.4 ± 1.7	(-)	(-)
SMP^2^Ag	50	50	14	938.4 ± 33.2 *	0.226	+58.1 ± 1.0 **	92.83 ± 9.15	11.40

Ag: *L*. *braziliensis* antigen; PLA: poly(*D*,*L*-lactic) acid; the hydrodynamic diameters and polydispersity (PdI) index were determined by dynamic light scattering (DLS); blank-SMP are particles without antigen. Encapsulation efficiency and payload (%) were calculated following Equations (1) and (2), respectively (method section). * Significantly different of blank-particles by unpaired *t*-test, (*n* = 9) * (*p* < 0.05), ** (*p* < 0.001).

### 3.2. Characteristics of the L. braziliensis-Loaded SMPs

The *Leishmania braziliensis* antigen (Ag) was associated with SMPs, and the SMPs are designated SMP^1^Ag and SMP^2^Ag, considering the same process and polymers for blank-SMP^1^ and blank-SMP^2^ preparations, respectively (Table 1). The size and zeta potential of the Ag-SMP^1^ were ~416 nm and +57 mV, whereas the SMP^2^Ag formulations were ~938 nm and +58 mV, respectively. As indicated in Table 1, at fixed concentrations of chitosan in the SMP^1^ (1%) and SMP^2^ (5%) formulations (Table 1), the association of negatively charged antigens significantly alters the SMP zeta potential by approximately 10 mV in both formulations (*p* < 0.001), probably by partially neutralizing the positive charge of chitosan on the surface of the particles. As the polydispersion indices (PdI) of populations of SMP^1^ particles indicate broad size distributions (PdI > 0.3), antigen association does not significantly change sizes. However, in blank-SMP^2^ with narrow size distributions (PdI < 0.3), sizes increase significantly with Ag association, as schematically represented in Figure 1.

The colloidal suspensions were able to encapsulate 69–93% of the antigen *L. braziliensis* (Table 1). The particles were able to load 28.5 g Ag/100 g and 11.4 g Ag/100 g of particles for SMP^1^Ag and SMP^2^Ag, respectively; however, the larger the particle size, the greater the encapsulation efficiency.

### 3.3. Kinetics of Cellular Infiltration in Mice Skin after Sensitization with SMPs

The inflammatory profile, toxicity and kinetics of cell migration induced by SMP^1^ and SMP^2^ in the inoculum site were evaluated at 1, 12, 24, 48 and 72 h after administration. All SMPs were able to promote cell recruitment to the site without tissue damage (Figure 1). At 1 h after inoculation, a discrete cellular infiltrate and vascular changes in the dermis and epidermis were observed in all samples (SMP^1^ and SMP^2^) when compared to the control (Figure 2A,F,K). At 12 h, a discrete polymorphonuclear inflammatory infiltrate located in the reticular dermis and hypodermis, edema, and spacing of collagen fiber regions were observed in the group treated with SMP^1^, but only in comparison to the control (Figure 2B,G). The inflammatory infiltrate of the SMP^2^ group at 12 h showed a diffuse distribution and mixed profile composed of polymorphonuclear cells and macrophages in the region of the papillary and reticular dermis and hypodermis with hyperemia when compared to the control (Figure 2B,L). At 24 h after inoculation, all findings were repeated and showed a progressive increase in the characteristics (Figure 2C,H,M). At 48 h, the SMP^1^ and SMP^2^ groups showed a diffuse infiltrate ranging from moderate to intense throughout the dermal compartment, predominantly composed of polymorphonuclear cells and macrophages. In addition, the presence of some lymphocytes in the subcutaneous region was observed in the SMP^2^ group (Figure 2D,I,N). At 72 h, a decrease in the inflammatory infiltrate with an early resolution process in both the SMP^1^ and SMP^2^ groups was observed (Figure 2E,J,O). The resolution process introduces features such as histological restoration profile of the dermis, fibroblast proliferation and restoration of skin appendages (hair follicles and glands).

A typical cellular infiltrate observed in the skin is illustrated in Figure 2I,N, with photomicrographs showing the cell recruitment within 48 h after blank-SMP inoculation. The cellular infiltrate was mainly composed of neutrophils, macrophages and lymphocytes. These data show the ability of blank-SMPs to promote local inflammation, which might be important for the start of the innate and acquired immune responses. The local inflammatory reaction did not induce macroscopic ulceration.

### 3.4. Cytokine Production in Mice Skin after Blank-SMPs Sensitization

Regarding IL-6 production after sensitization with different SMPs, the CT group showed an increase at 12 h compared to 1 and 72 h (Figure 3A). The SMP^1^ group presented a significant increase at 12 h compared to 1, 24 and 48 h and a decrease at 24 and 48 h when compared to 1 h (Figure 3A). For the SMP^2^ group, a significant increase at 12 h compared to 72 h was observed. By drawing a comparison between groups, significantly higher levels of IL-6 in SMP^1^ compared with the CT group were detected 1 h after inoculation with the particles (Figure 3A). 

Interestingly, TNF-α levels were detected only in the SMPs groups (Figure 3B), with production in SMP^1^ at 12 h, in addition to SMP^2^ at 1 h, 12 h and 72 h. 

### 3.5. Proinflammatory Soluble Protein in Mice Skin at Different Time Points after Sensitization with Blank-SMPs

The profile of soluble protein was obtained from homogenized skin tissue after blank-SMP stimulation. MPO and NAG were measured by biochemical assay. For the MPO, similar results were demonstrated among all the analyzed groups (Figure 4A). However, when a comparison was made between the groups, a low level of MPO at 72 h was observed for SMP^1^ when compared to the CT group (Figure 4A). Concerning the NAG enzyme, although no differences were identified in the control group, in the SMP^1^ group, a significant increase at 1, 24 and 48 h, when compared to 72 h (*p* < 0.05), was observed. A significant increase in NAG levels for the SMP^2^ group was observed at 12 h when compared to 24 and 72 h. This group also showed higher levels of NAG at 12 h when compared with the group treated with SMP^1^ (*p* < 0.05) (Figure 4B).

### 3.6. Immunophenotyping of Circulating Leukocytes after Sensitization with Blank- and Antigen-Loaded SMPs

Following the evaluation of the compartmentalized immune response, the systemic response was investigated through ex vivo immunophenotypic analyses of the peripheral blood cells profile by flow cytometry 14 days after inoculation with blank- or antigen-loaded SMPs (Figure 5). The innate immune response in the peripheral blood showed a decrease in monocytes (CD14^+^) in the SMP^1^ group when compared with CT and SMP^1^Ag (*p* < 0.05) (Figure 5A). A reduction in the circulating NK cells (CD3^−^CD49^+^) was observed in all groups when compared with CT (*p* < 0.05). Furthermore, the blank or encapsulated SMP^1^ showed increased levels of NK cells when compared with the blank or encapsulated SMP^2^ (Figure 5B). In addition, the LBSap group showed an increase in circulating NK cells when compared to the blank or encapsulated SMPs (*p* < 0.05).

The analyses of acquired immune response in peripheral blood showed a decrease in CD3^+^CD4^+^ T lymphocytes in SMP^2^ when compared to CT, SMP^1^ and SMP^2^Ag (Figure 5C). Furthermore, when evaluating CD3^+^CD8^+^ T lymphocytes, the SMP^2^ group presented a lower percentage when compared to CT, SMP^1^, SMP^1^Ag and SMP^2^Ag (*p* < 0.05) (Figure 5D). Relative to the CD19^+^ B lymphocytes, it was noted that all groups presented a decrease in circulating B lymphocytes when compared to CT (*p* < 0.05), with the exception of SMP^2^Ag. Moreover, SMP^1^ showed a drop in the percentage of CD19^+^ B lymphocytes when compared to SMP^1^Ag, SMP^2^ and SMP^2^Ag (*p* < 0.05). Furthermore, SMP^2^Ag showed higher levels of CD19^+^ in relation to SMP^2^ (*p* < 0.05) (Figure 5E).

## 4. Discussion

The development of an effective vaccine depends not only on the antigen but also on the system that promotes increased immunogenicity, such as the adjuvant and delivery system. The co-administration of antigens with particulate adjuvants can improve the immune response of vaccines, avoid rapid enzymatic degradation of protein and peptides and modulate the antigen release [37]. Herein, two chitosan-coated polymeric submicrometric particle (SMPs) sets were selected and evaluated, aimed at conducting an analysis of the in situ toxicity and inflammatory profile. A benefit of the use of chitosan is a more sustained release of the antigen and the stimulation of the immune system provided by its polycationic nature and gelling properties of this polysaccharide in the particulate form, as reviewed [38]. Moreover, chitosan-coated PLA SMPs were used as an adjuvant and delivery system to load *L. braziliensis* antigens that had previously been shown to trigger intense immunogenicity [29] and protection profile against canine visceral leishmaniasis [30,31].

Chitosan-coated PLA SMPs prepared by the modified nanoprecipitation method have been successfully employed to provide two sizes of SMP-loading crude antigens of *L. braziliensis,* SMP^1^Ag and SMP^2^Ag with mean sizes of 416 nm and 938 nm, respectively, and a similar surface charge of approximately +57 mV. Both were chosen for the in vivo studies. All the formulations showed positive zeta potential values, indicating that polycationic chitosan polymer is located at the SMP surface, as schematized in Figure 1.

The sizes of particles were influenced by the percentage of polymers added in the preparation, 1 or 5%. Antigen loading has no effect on the polydispersion indexes of the SMPAg compared with blank-SMPs. As SMP^2^Ag has bigger sizes, Ag has a higher effect on the smaller surface area. A negatively charged antigen association with SMP reduces in modulus blank-SMP positive zeta potential. High zeta potential values boost the colloidal stability of formulations, thus avoiding particle aggregation [39]. Concerning the ability to transport the antigen (payload), the smaller particles (SMP^1^) with high surface areas are able to associate more antigen, probably because the antigen is adsorbed at the hydrophilic chitosan chains located at the surface of the particles. Oppositely, encapsulation efficiency increases with the increasing mass of polymers, particularly the chitosan, indicating its effect on protein adsorption at the nanoparticle’s surface. Furthermore, the polycationic nature of chitosan is usually more efficient in eliciting an appropriate immune response to an adjuvant effect than negatively charged nanocarriers [40,41]. Similar values of association of recombinant *Leishmania* superoxide dismutase with chitosan-coated nanoparticles of 22–75% were also observed for nanoparticles of 59–850 nm mean sizes with positively charged surfaces [11].

Considering the potential formulations, an in vivo test using empty particles (blank-SMPs) was conducted to characterize their potential role as adjuvants and their use in vaccine formulations.

An early and effective immune response triggered by adjuvants leads to improvements in the immunogenicity and efficacy of vaccines since there is an induction of the innate immune response that drives the acquired immune response [42,43,44]. As such, adjuvants are essential to increasing the immunogenicity of local inflammation through the recruitment and activation of lymphocytes and prolonging the persistence of the antigen [45,46,47]. The present study offers insight into the innate immune response in mice after a single dose of SMPs. Skin samples were collected in this study to analyze the inflammatory profile after inoculation with SMP particles once they were administered intradermally. Moreover, the skin is the first point of contact between the vertebrate host and the parasite, in addition to being the first point of contact with the vector’s immunomodulatory molecules. In this sense, it is extremely important to study the initial events triggered by vaccination in this compartment [48]. The qualitative cellular infiltrate showed that all SMPs evaluated caused cell infiltration at the inoculation site. Overall, these results demonstrated that SMP^1^ presented an important reduction of neutrophilic activity (NAG) at 72 h (compared to the CT group), and SMP^2^ demonstrated a progressive reduction in the NAG levels during the kinetics follow-up (especially by comparison between 12 h and 72 h). Importantly, while NAG levels were similar in the CT group, SMP^1^ and SMP^2^ exhibited a reduction in neutrophil infiltration at the final time points. The ability to induce a local inflammatory response is an important feature and is shared by most of the known adjuvants, with these data being consistent with the results from other authors [25,49]. Despite the local inflammatory reaction having been observed by optical microscopy, no macroscopical ulcerations were observed. It has been suggested that the immune response is proportionally related to the tissue damage elicited by the adjuvants. Therefore, new adjuvants should mimic danger signals, preferably with minimal injury to healthy tissue [49,50].

De Moura et al. (2009) reported that the dosage of enzymes produced by neutrophils (MPO) or macrophages (NAG) is a technique used as a surrogate of the infiltration rate of these cells at the inflammatory sites [36]. The present study’s data revealed a similar migration of neutrophils and macrophages to the inoculation site, and the presence of these cells was more persistent in the groups sensitized with SMP^1^ and SMP^2^ at 24 and 48 h. The increased MPO levels indicate a neutrophil activation profile, which has been observed in all groups and at all times; however, maximum activity was observed at 48 h. Moreover, the SMP^1^ group showed a lower enzyme expression after 72 h when compared to the CT group, indicating inflammatory control. Calabro et al. (2011) hypothesized that MF59 adjuvant action is based on innate immune cell recruitment at the inoculation site, with neutrophils being the first and most abundant cell type [51]. Again, this study’s data are consistent with Calabro’s results [51]. Notably, neutrophils may play an important role as the vehicle for transporting the vaccine antigen into the draining lymph nodes for further processing and presentation, presumably by dendritic cells [52,53]. While the CT group did not present a significant difference in NAG levels, the groups SMP^1^ and SMP^2^ demonstrated detected levels at all scheduled times. Moreover, the macrophage analysis by MPO activity revealed similar enzymatic levels in all the analyzed groups. Macrophages act as the first line of the immune system’s defense, producing diverse cytokines, such as IL-1, IL-6, IL-8 and TNF-α, which mobilize innate immune reactions and signal T cells to initiate specific responses against intracellular and extracellular pathogens [54]. By using different adjuvants, Vitoriano-Souza et al. (2012) reported the recruitment of neutrophils, macrophages and lymphocytes a few hours after applying a single dose of an adjuvant, which is closer to the results obtained in this experiment based on the histopathological analysis [55].

In addition to the aforementioned analyses, the cytokine profile was evaluated to better understand the early events related to the immune response to SMPs. High levels of IL-6 at 12 h were observed in all groups, followed by a reduction in these levels after this time. Regarding TNF- α, a peak at 12 h for SMP^1^ was observed and detectable at low levels for SMP^2^. Gutiérrez et al. (2015) described that an effective immune response against *Leishmania* is directly related to the nanoparticles’ physicochemical properties when used as adjuvants in a vaccine [14]. The presence of proinflammatory cytokines at the first time of inoculation is necessary for immune response polarization and enhancing the vaccine efficacy [52,56]. Tafaghodi et al. (2010) demonstrated that empty PLGA nanospheres induced protective immune responses as a vaccine delivery system in contrast to *Quillaja saponins,* which had the opposite effect as an adjuvant [57]. Lage et al. (2020) demonstrated that a *Leishmania* chimera antigen, when associated with nanoliposomes, could induce a type I response and protection against *L. infantum* infection in BALB/c mice [58]. Despite the induction of IL-6 production and MPO in the inoculation site of the control group, it is possible to hypothesize that the mechanical stress on the extracellular matrix and cells can trigger slight transient inflammation, as previously reported [59].

Overall, these findings suggest that chitosan-coated PLA particles can induce a detectable level of proinflammatory cytokines (TNF-α) and soluble proteins (NAG), providing potential stimulus for cell migration. These promising data guided the analysis of *L. braziliensis* loaded SMPs performance, resulting in the SMP^1^Ag and SMP^2^Ag formulations. These results show that particles containing *Leishmania* antigen do not significantly alter the physicochemical characteristics when compared to the unloaded SMPs. Some studies have already shown that the *L. braziliensis* antigen, when in association with saponin as an adjuvant (LBSap vaccine), triggers a highly efficacious immune response against visceral leishmaniasis similar to commercial vaccines, validating their potential as a promising vaccine candidate for CVL control [28,31,59]. That being so, SMPs loaded with *L. braziliensis* antigens might enhance its immunogenicity pattern. SMP^1^Ag and SMP^2^Ag showed zeta potential values less positive than the unloaded particles. This suggests an electrostatic interaction between the antigen (negatively charged) and the positive surface charges of SMPs. The antigens of *L. braziliensis* have a net negative zeta potential. In fact, the two formulations (SMP^1^ and SMP^2^) were effective in the antigen encapsulation process since 69–93% of the antigen was encapsulated, indicating that the SMPs produced from this polymeric combination have a high ability to transport these selected *Leishmania* antigens.

The SMPs loaded with *L. braziliensis* antigens were preliminarily analyzed as a potential vaccinal candidate. The systemic immune response of SMPs was assessed by ex vivo immunophenotypic analyses in the peripheral blood after 14 days of the third immunization. When compared to the blank-SMPs, a significant increase in the monocytes and B-lymphocytes was observed in the SMP^1^Ag group. In addition, the SMP^2^Ag group showed a significant increase in the CD3^+^CD4^+^ and CD3^+^CD8^+^ T-lymphocytes and B-lymphocytes when compared to the SMP^2^ group. Several studies have shown that memory T cells are critical to inducing protection against infections in the long term [28,30,31,60]. Furthermore, it has been reported that LBSap-vaccinated dogs and mice showed high counts of T cells and B cells, similar to the results here reported [28,29,30,31]. These preliminary immunogenicity data suggest the SMP formulations were able to interfere with the levels of NK (CD3^−^CD49^+^) cells in all SMPs groups, in addition to impairment in the SMP^2^ group of T cells subsets (CD3^+^CD4^+^) and CD3^+^CD8^+^) and B cells (CD19^+^).

## 5. Conclusions

The data presented here suggest that chitosan-PLA-based formulations present biocompatibility, induce an in situ (detectable levels of NAG and TNF-α) and systemic immune response [reduction in NK (CD3^−^CD49^+^) cells in all SMPs groups; reduction in the SMP^2^ group of T cells subsets (CD3^+^CD4^+^) and CD3^+^CD8^+^) and B cells (CD19^+^)] with a controlled inflammation profile. In fact, the two potential formulations that could be used to develop a vaccine against VL stimulate the early events of an innate immune response at the inoculation site, suggesting their ability to enhance the immunogenicity of co-administered antigens. However, in view of the results, it is not possible to highlight which of the two tested polymeric particles presents the best vaccine performance since additional studies focusing on the protection rate against *L. infantum* infection are required. The encapsulation of *Leishmania* antigens in SMPs has demonstrated their potential formulation in VL vaccine candidates.

## Figures and Tables

**Figure 1 vaccines-11-01309-f001:**
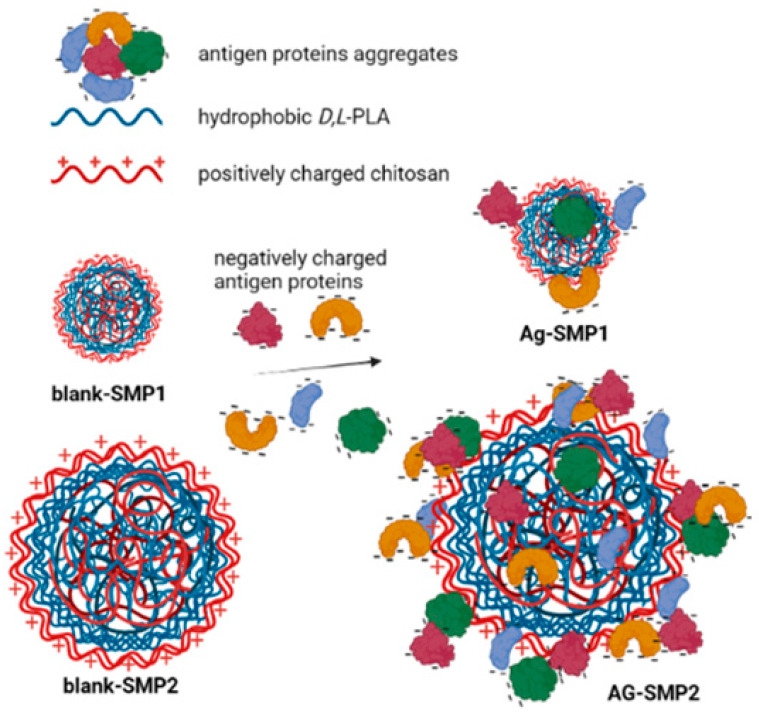
Schematic representation of submicrometric particles SMP^1^ and SMP^2^, before (blank-SMP) and after antigen (Ag) association (SMPAg). Antigen aggregates show negatively charged surfaces. The association of proteins (Ag) induces partial neutralization of chitosan positive charges at the SMP surface, altering zeta potential. The larger SMP^2^ particles have more capability to associate Ag than smaller SMP^1^ (higher EE%). Ag association induces more effect on sizes of SMP^2^Ag (Figure created with BioRender.com).

**Figure 2 vaccines-11-01309-f002:**
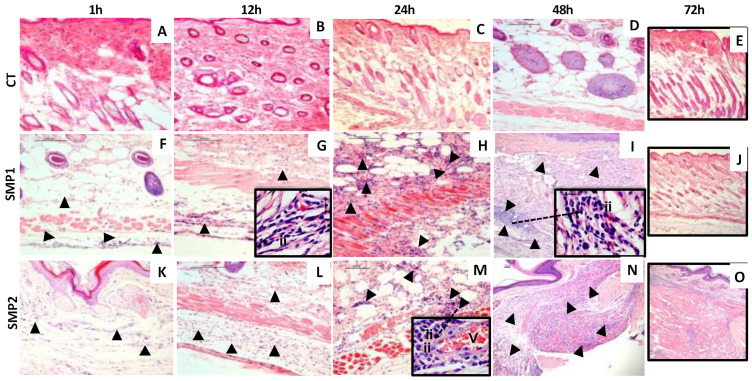
Histological analysis in the skin fragments obtained after 1, 12, 24, 48 and 72 h of the inoculum. Histological section photomicrographs (5 µm, HE) of skin fragments from mice inoculated with saline (CT–Figure (**A**–**E**)), blank-SMP^1^ (SMP^1^-Figure (**F**–**J**)) and blank-SMP^2^ (SMP^2^-Figure (**K**–**O**)) in different time points. The photomicrographs are representative of leukocyte migration kinetics throughout the experimental period with progressive and intense inflammatory infiltrate in the reticular dermis and resolution at 72 h. V: vascular sprouts; dotted arrows: ii: inflammatory infiltrate. Bar = 25 μm (10×), 50 μm (20×). Insert = 40× zoom.

**Figure 3 vaccines-11-01309-f003:**
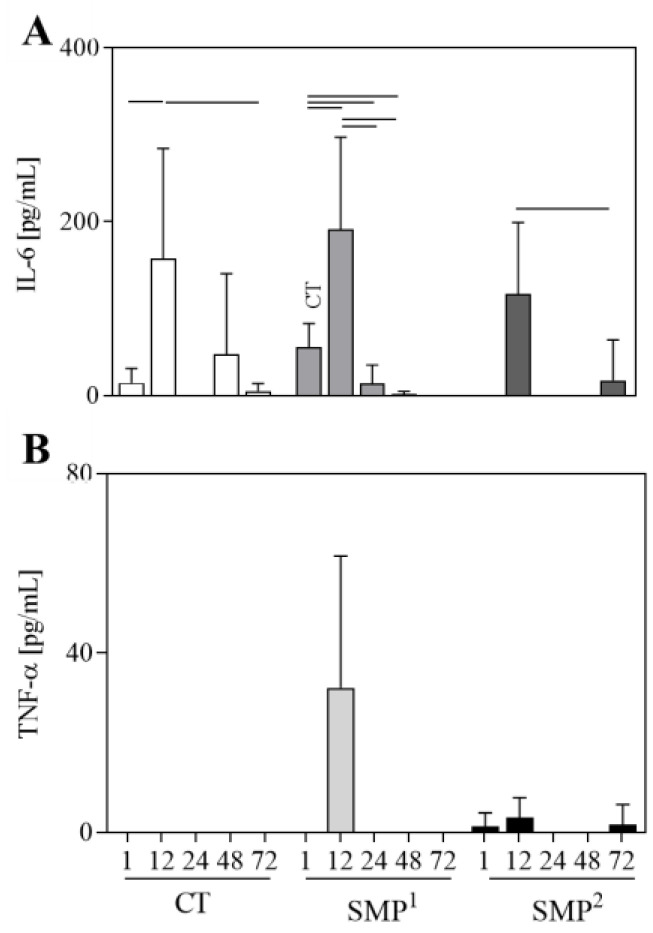
IL-6 (**A**) and TNF-α (**B**) levels in different times (1, 12, 24, 48 and 72 h) from macerated skin supernatant after immunization with saline (CT), blank- SMP^1^ and blank-SMP^2^. The results are shown as a column bar graph and represent the mean and standard deviation. Significant differences (*p* < 0.05) in the same group are represented by connecting lines, and the difference in relation to the control group is represented by CT.

**Figure 4 vaccines-11-01309-f004:**
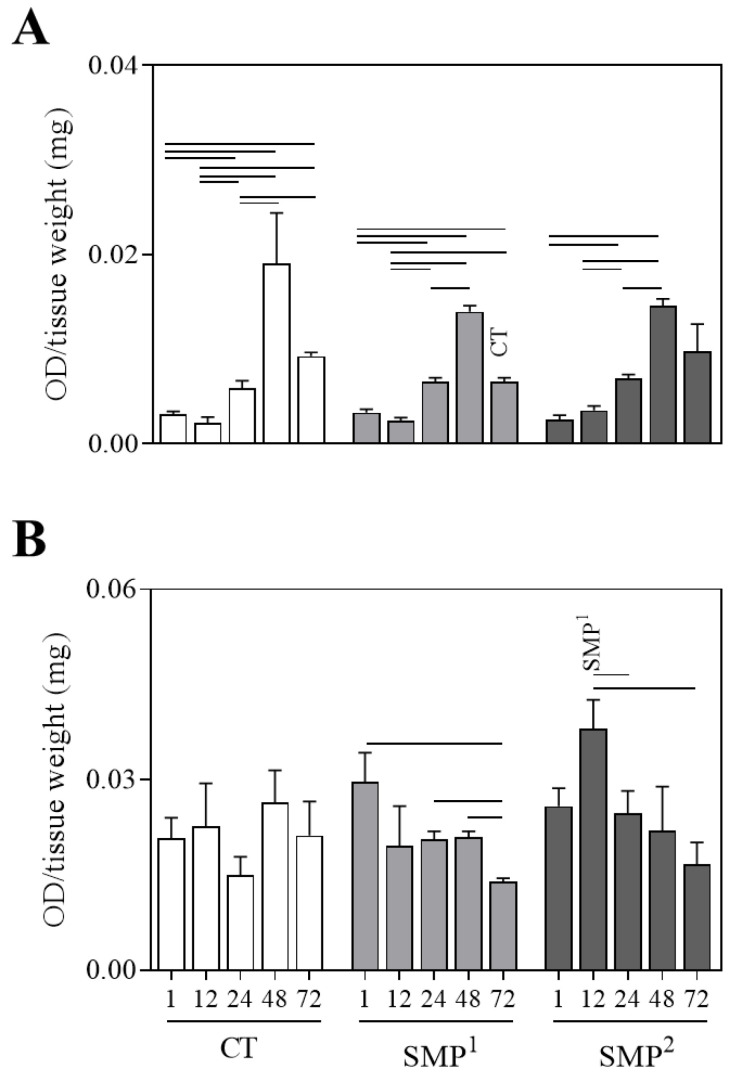
Protein dosage in macerated skin obtained from the inoculation site. Evaluation of MPO (**A**) and NAG (**B**) at different time periods (1, 12, 24, 48 and 72 h) from macerated skin supernatant after immunization with saline (CT), empty submicrometer particles 1 (blank-SMP^1^) and submicrometer particles 2 (blank-SMP^2^). The results are shown as a column bar graph and represents the mean and standard deviation. Significant differences (*p* < 0.05) in the same group are represented by connecting lines, while differences in relation to the control group are represented by CT and differences in relation to the SMP^1^ group are represented by SMP^1^.

**Figure 5 vaccines-11-01309-f005:**
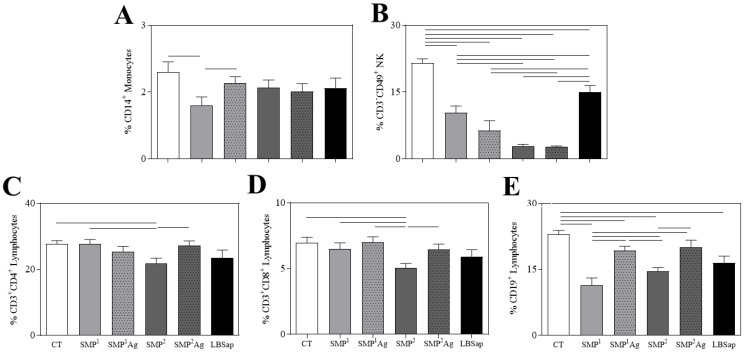
Innate and adaptative immune profile of circulating monocytes (CD14^+^− (**A**)), NK cells (CD3^−^CD49b^+^−(**B**)), lymphocytes T (CD3^+^CD4^+^− (**C**); or CD3^+^CD8^+^− (**D**)) and lymphocyte B (CD19^+^− (**E**)), determined 14 days after immunization: CT (control); SMP^1^ (empty submicrometer particles 1); SMP^1^Ag (submicrometer particles 1 with loaded with *L. braziliensis* antigens); SMP^2^ (empty submicrometer particles 2); SMP^2^Ag (submicrometer particles 2 loaded with *L. braziliensis* antigens); and LBSap vaccine (*L. braziliensis* antigens plus saponin as adjuvant). The results are shown as a column bar graph and represents the mean and standard deviation. Significant differences (*p* < 0.05) are represented by connecting lines.

## Data Availability

The data that support the findings of this study are available from the corresponding author, [RCG], upon reasonable request.

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
