# Peer review of "Polymeric Delivery Systems as a Potential Vaccine against Visceral Leishmaniasis: Formulation Development and Immunogenicity"

_vaccines, 2023, doi:10.3390/vaccines11081309_

Round 1

Reviewer 1 Report

In this manuscript, da Silva et al assessed the effect of chitosan coated poly(D,L-lactic) acid submicrometric particles and encapsulation of L. braziliensis antigens in these particles, on immune function. Overall, there is a strong need to develop new immunization strategies for leishmaniasis. However, this manuscript does not show much difference between the authors’ new formulation vs saline, in terms of immunogenicity. I believe this is an appropriate venue for reporting negative results; however, these should be clearly reported as such. Instead, authors make statements that are either not supported by the data or missing caveats about similar findings in saline groups. A more accurate description of their results would be that these formulations provide no additional benefits. A detailed description of these issues is as follows:

Major issues:

1.     These statements need statistical support: lines 282-283: “The antigen encapsulation does not significantly alter the overall size of the colloidal suspension.” And lines 285-286: “The adsorption of antigen influences the zeta potential and reduces the positive values of the zeta potential of particles”, as well as lines 422-424.

2.     Not all the inflammatory infiltration figures are convincing. Some of the structures indicated by arrows in panel 1G look the same as in panel 1B. Some of the structures in 1H and 1I top are also very small. This figure may be easier to interpret if no-injection controls are also included as reference to enable evaluation of what “normal” skin looks like. A higher resolution figure should also be provided.

3.     Using single histological examples in Figure 1 is insufficient, especially since some of the purported infiltrates are so small. Authors should provide quantitative metrics, such as measurement of the surface area of inflammatory infiltrates across experimental groups. This is particularly important since Figure 3 quantitative analysis shows few significant differences compared to saline.

4.     Line 424: this statement needs data to support it: “Ag seems to be located preferentially at the particle surfaces”.

5.     Lines 338-342: these statements do not appear to be supported by statistical analysis. If these are non-significant, they should not be discussed.

6.     There are multiple over-statements in the abstract, introduction and discussion, as follows:

a.     Lines 34-35: “they induced a selective migration of neutrophils and macrophages, with the inflammatory response being reduced at 72 h in all groups.” This statement is incorrect, since basically no significant differences to saline were observed.

b.     Line 35: “The SMPs led to NAG and MPO production”. This is incorrect, since no difference to control saline was observed in figure 3. Instead, this is likely to be a consequence of the needle disrupting the tissue.

c.     Line 38-39: “The immunogenicity analysis by flow cytometry demonstrated an increase in monocytes and B-lymphocytes in the SMP1Ag group and an increase in T-lymphocytes and B-lymphocytes in SMP2Ag, as compared to blanked SMPs.” While this statement is technically correct, presenting it on its own is misleading, since none of the SMPAg groups out-performed control saline.

d.     Line 98-99: “In addition, the safety and toxicity of different submicrometric particles were analyzed”. In fact, no safety or toxicity assessment are presented in the manuscript. Sections marked “data not shown” should instead provide the data, either as a main figure or as a supplemental figure.

e.     Lines 40-43: “The resulting data demonstrate that the chitosan coated polymeric sub-microparticle formulations stimulate the early events of innate immune response, suggesting their ability to increase the immunogenicity of co-administered Leishmania antigens.” Given that there was effectively no benefit compared to control saline in figures 2, 3 and 4, this statement suggests that instead other aspects of the injection (such as needle damage) are instead responsible for any observed effect.

f.      Lines 444-446: “Overall, these results demonstrated that SMP1 and SMP2 induce a quick local recruitment of inflammatory cells in situ and this response is maintained up to 48 h after inoculation followed by resolution of inflammation.” The lack of significant difference to saline in Figure 3 should be clearly stated here, otherwise this statement seems to imply a superiority that is not actually experimentally demonstrated.

g.     Line 469-470: “The NAG levels were detected at all scheduled times, consistent with the observed macrophage infiltrate in the SMPs groups.” This is misleading if it is not qualified by the fact that similar responses were observed in the saline group.

h.     Lines 492-494: “Overall, these findings suggest that chitosan-coated PLA particles can induce an early production of pro-inflammatory cytokines and soluble proteins (NAG and MPO) with ensuing cell migration within a few hours after inoculation.”. This statement is misleading, since similar responses were also observed in the saline group and are thus not caused by the particles themselves.

i.       Lines 520-522: “These preliminary immunogenicity data suggest the SMP formulations were able to stimulate an immune adaptative response after immunization protocol.” This is overall incorrect, since the SMP formulations overall caused very similar responses to saline (or less of a response), in most quantitative assays.

j.       Line 525-526: “induce an in situ and systemic immune response with a controlled inflammation profile.” This statement is also misleading, since overall no differences to saline were observed.

k.     Lines 529-531: “The encapsulation of Leishmania antigens in SMPs has demonstrated their potential to boost the immunogenicity of the VL vaccine candidates, thus indicating new perspectives for the development of new vaccine formulations against Leishmania infections.” This is likewise misleading, since LBSap produced either comparable or stronger responses in Figure 4.

7.     Given the lack of additional benefit observed for the different formulations compared to saline, this manuscript should be revised throughout to clearly demonstrate that there was little to no observable benefit to these formulations.

Minor issues:

1.     While I appreciate that the authors have previously shown that their L. braziliensis vaccine protects against L. infantum, nevertheless this vaccine strategy is based on a mucocutaneous Leishmania species, and so the emphasis on VL in the title, keywords and introduction is a bit misleading, especially since no study of protective efficacy was performed here.

2.     Line 203-204: typo? “12.000 glucosaminidase × g”

3.     Lines 515-517: “In addition, the SMP2 group showed a significant increase in the CD3+CD4+ and CD3+CD8+ T-lymphocytes and B-lymphocytes”. Reference point should be clearly mentioned (SMP2-Ag vs naked SMP2).

Author Response

Comments and Suggestions for Authors

In this manuscript, da Silva et al assessed the effect of chitosan coated poly(D,L-lactic) acid submicrometric particles and encapsulation of L. braziliensis antigens in these particles, on immune function. Overall, there is a strong need to develop new immunization strategies for leishmaniasis. However, this manuscript does not show much difference between the authors’ new formulation vs saline, in terms of immunogenicity. I believe this is an appropriate venue for reporting negative results; however, these should be clearly reported as such. Instead, authors make statements that are either not supported by the data or missing caveats about similar findings in saline groups. A more accurate description of their results would be that these formulations provide no additional benefits. A detailed description of these issues is as follows:

Major issues:

  1. These statements need statistical support: lines 282-283: “The antigen encapsulation does not significantly alter the overall size of the colloidal suspension.” And lines 285-286: “The adsorption of antigen influences the zeta potential and reduces the positive values of the zeta potential of particles”, as well as lines 422-424.

Thanks for highlighting this point. We add the statistical analysis in Table 1 as well as standard deviations of the 9 size measurements concerning sizes.

We modify the text in the manuscript to be clear.

As indicated in this Table, there is no significant difference in mean size between blank-SMP1 and antigen-loaded SMP1. This is likely because both formulations are polydisperse in size. In contrast, there is a difference in mean size between the larger particle formulations, SMP2, after antigen loading (p < 0.05). Antigen adsorption produces larger average sizes and partially neutralizes the positive charges related to the chitosan polymer. It has the effect of reducing the zeta potential values in modulus. This size increase is in line with the better association (higher %EE) of antigens on the larger polymeric particles (SMP2). Antigen association has the same effect on the zeta potential of smaller NPs indicating that negatively charged antigens are able to partially neutralize the positive charges provided by the chitosan coating on both types of NP.

  1. Not all the inflammatory infiltration figures are convincing. Some of the structures indicated by arrows in panel 1G look the same as in panel 1B. Some of the structures in 1H and 1I top are also very small. This figure may be easier to interpret if no-injection controls are also included as reference to enable evaluation of what “normal” skin looks like. A higher resolution figure should also be provided.

We are thankful for this comment and we agree with the Reviewer #1. The representative Figure has been replaced by higher-resolution images, and we have added a 40X magnification for the identification of inflammatory infiltrates.

  1. Using single histological examples in Figure 1 is insufficient, especially since some of the purported infiltrates are so small. Authors should provide quantitative metrics, such as measurement of the surface area of inflammatory infiltrates across experimental groups. This is particularly important since Figure 3 quantitative analysis shows few significant differences compared to saline.

The enzymatic assays of NAG and MPO correspond to indirect quantitative methods presenting direct relationship with activated cells present in the inflammatory focus, thus providing reliable data regarding the profile of the innate immune response, which was our main objective in this study. In fact, some previously reports have been used NAG and MPO as enzymatic biomarkers to assess the infiltration of macrophages and neutrophils, respectively (Lanna et al., 2020 - DOI: 10.3389/fbioe.2020.538203; Lima et al., 2014 - DOI: 10.1186/1472-6882-14-177; Oliveira et al., 2015 - DOI: 10.1002/jps.24569; de Moura et al., 2011 - DOI: 10.1093/ecam/nep197).

  1. Line 424: this statement needs data to support it: “Ag seems to be located preferentially at the particle surfaces”.

We agree and we remove this statement from the text. Zeta potential is one of the indicators of this location. The antigens aggregates are negatively charged (Table 1 Ag aggregates) and after association with positively charged chitosan coated-NP, these antigens partially neutralize NP surface charges inducing significant differences in zeta potentials (p < 0.001). We add Figure 1 to clarify the schematic representation of this point.

Figure 1. Schematic representation of submicrometric particles SMP1 and SMP2, before (blank-SMP) and after antigen (Ag) association (SMPAg). Antigen aggregates show negatively charged surfaces. The association of proteins (Ag) induces partial neutralization of chitosan positive charges at the SMP surface, altering zeta potential. The larger SMP2 particles have more capability to associate Ag than smaller SMP1 (higher EE%). Ag association induces more effect on sizes of SMP2Ag (Figure created with BioRender.com).

  1. Lines 338-342: these statements do not appear to be supported by statistical analysis. If these are non-significant, they should not be discussed.

We are thankful for this comment. In fact, we corrected the sentence that appears in the revised manuscript as follows:

“(…) Interestingly, TNF-α levels were detected only in the SMPs groups (Figure 2B), with production in SMP1 at 12h, in addition to SMP2 at 1h, 12h and 72h. (…)”.

  1. There are multiple over-statements in the abstract, introduction and discussion, as follows:
  2. Lines 34-35: “they induced a selective migration of neutrophils and macrophages, with the inflammatory response being reduced at 72 h in all groups.” This statement is incorrect, since basically no significant differences to saline were observed.

We have made appropriated changes in the Abstract to clarify this issue and the text appeared in the revised manuscript as follows:

“(…) In addition, besides the myeloperoxidase (MPO) activity has demonstrated similar results among the analyzed groups, a progressive reduction in the levels of N-acetyl--D-glucosaminidase (NAG), until 72h were observed for SMPs. While IL-6 levels were similar among all analyzed groups along the kinetics, only the SMPs groups had detectable levels of TNF-α. (…)”.

  1. Line 35: “The SMPs led to NAG and MPO production”. This is incorrect, since no difference to control saline was observed in figure 3. Instead, this is likely to be a consequence of the needle disrupting the tissue.

In fact, we have made changes in the revised version of the manuscript, and this sentence appears as follows:

“(…) While IL-6 levels were similar among all analyzed groups along the kinetics, only the SMPs groups had detectable levels of TNF-α (…)”.

  1. Line 38-39: “The immunogenicity analysis by flow cytometry demonstrated an increase in monocytes and B-lymphocytes in the SMP1Ag group and an increase in T-lymphocytes and B-lymphocytes in SMP2Ag, as compared to blanked SMPs.” While this statement is technically correct, presenting it on its own is misleading, since none of the SMPAg groups out-performed control saline.

We agree with the reviewer, and the sentence was corrected highlighting the differences with Control group, as follows:

“(…) The immunogenicity analysis by flow cytometry demonstrated a reduction in NK (CD3-CD49+) cells in all SMPs groups, in addition to impairment in the T-cells subsets (CD3+CD4+) and CD3+CD8+) and B-cells (CD19+) of SMP2 group. (…)”.

  1. Line 98-99: “In addition, the safety and toxicity of different submicrometric particles were analyzed”. In fact, no safety or toxicity assessment are presented in the manuscript. Sections marked “data not shown” should instead provide the data, either as a main figure or as a supplemental figure.

We removed the sentence from the text about renal and hepatic functions that mentioned “data not shown”.

  1. Lines 40-43: “The resulting data demonstrate that the chitosan coated polymeric sub-microparticle formulations stimulate the early events of innate immune response, suggesting their ability to increase the immunogenicity of co-administered Leishmania antigens.” Given that there was effectively no benefit compared to control saline in figures 2, 3 and 4, this statement suggests that instead other aspects of the injection (such as needle damage) are instead responsible for any observed effect.

We are extremely thankful to the Reviewer #1 for the comments of the abstract that was changed highlighting the differences with Control group. We have made appropriated changes in the final sentence of the abstract that appears as follows:

“(…) In addition, besides the myeloperoxidase (MPO) activity has demonstrated similar results among the analyzed groups, a progressive reduction in the levels of N-acetyl-b-D-glucosaminidase (NAG), until 72h were observed for SMPs. (…)”.

  1. Lines 444-446: “Overall, these results demonstrated that SMP1 and SMP2 induce a quick local recruitment of inflammatory cells in situ and this response is maintained up to 48 h after inoculation followed by resolution of inflammation.” The lack of significant difference to saline in Figure 3 should be clearly stated here, otherwise this statement seems to imply a superiority that is not actually experimentally demonstrated.

We believe that after the replacement of Fig. 2 and the clarification of Fig. 3, this issue was clarified since the production of TNF-alpha was observed just in SMP groups, as described in the manuscript. Moreover, we corrected this sentence according to the results and the revised sentence appeared as follows:

“(…) Overall, these results demonstrated that SMP1 presented an important reduction of neutrophilic activity (NAG) at 72h (compared to CT group) and SMP2 demonstrated a progressive reduction in the NAG levels during the kinetics follow up (specially by comparison between 12h and 72h). Importantly, while NAG levels were similar in the CT group, SMP1 and SMP2 exhibited a reduction in neutrophil infiltration at the final time points. (…)”

  1. Line 469-470: “The NAG levels were detected at all scheduled times, consistent with the observed macrophage infiltrate in the SMPs groups.” This is misleading if it is not qualified by the fact that similar responses were observed in the saline group.

We have made appropriate changes in this sentence that appears as follows:

“(…) While the CT group did not present a significant difference in NAG levels, the groups SMP1 and SMP2 demonstrated detected levels at all scheduled times. Moreover, the macrophage analysis by MPO activity revealed similar enzymatic levels in all analyzed groups. (…)”.

  1. Lines 492-494: “Overall, these findings suggest that chitosan-coated PLA particles can induce an early production of pro-inflammatory cytokines and soluble proteins (NAG and MPO) with ensuing cell migration within a few hours after inoculation.”. This statement is misleading, since similar responses were also observed in the saline group and are thus not caused by the particles themselves.

We agree with Reviewer #1 and this sentence was clarified as follows:

“(…) Overall, these findings suggest that chitosan-coated PLA particles can induce an detectable levels of pro-inflammatory cytokines (TNF-a) and soluble proteins (NAG) providing potential stimulus for cell migration (…)”.

  1. Lines 520-522: “These preliminary immunogenicity data suggest the SMP formulations were able to stimulate an immune adaptative response after immunization protocol.” This is overall incorrect, since the SMP formulations overall caused very similar responses to saline (or less of a response), in most quantitative assays.

We agree with Reviewer #1. We have made changes in this sentence and the revised version of the manuscript appears as follows:

“(…) These preliminary immunogenicity data suggest the SMP formulations were able to interfere with the levels of NK (CD3-CD49+) cells in all SMPs groups, in addition to impairment in the SMP2 group of T-cells subsets (CD3+CD4+) and CD3+CD8+) and B-cells (CD19+). (…)”.

  1. Line 525-526: “induce an in situ and systemic immune response with a controlled inflammation profile.” This statement is also misleading, since overall no differences to saline were observed.

According the previous suggestions of the Reviewer #1, we have made changes in all the results. For this reason, we adapted this sentence that appears in the revised manuscript as follows:

“(…) The data presented here suggest that chitosan-PLA-based formulations present biocompatibility, induce an in situ (detectable levels of NAG and TNF-a) and systemic immune response [reduction in NK (CD3-CD49+) cells in all SMPs groups; reduction in the SMP2 group of T-cells subsets (CD3+CD4+) and CD3+CD8+) and B-cells (CD19+)] with a controlled inflammation profile. (…)”

  1. Lines 529-531: “The encapsulation of Leishmania antigens in SMPs has demonstrated their potential to boost the immunogenicity of the VL vaccine candidates, thus indicating new perspectives for the development of new vaccine formulations against Leishmania infections.” This is likewise misleading, since LBSap produced either comparable or stronger responses in Figure 4.

We agree with the Reviewer #1 comments. Considering all the changes performed in the revised version of the manuscript, we adapted the final sentence as follows:

“(…) The encapsulation of Leishmania antigens in SMPs has demonstrated their potential for formulation in VL vaccine candidates. (…)”.

  1. Given the lack of additional benefit observed for the different formulations compared to saline, this manuscript should be revised throughout to clearly demonstrate that there was little to no observable benefit to these formulations.

We are thankful to the Reviewer #1 comments that increase the quality of the results and the discussion section. We change the focus of the results demonstrating the differences with the Control group that become the manuscript more reliable with the data obtained.

Minor issues:

  1. While I appreciate that the authors have previously shown that their L. braziliensis vaccine protects against L. infantum, nevertheless this vaccine strategy is based on a mucocutaneous Leishmania species, and so the emphasis on VL in the title, keywords and introduction is a bit misleading, especially since no study of protective efficacy was performed here.

We appreciate Reviewer #1's comment. In fact, our research group have been developing a heterologous vaccine composed by L. braziliensis antigens to protect against the L. infantum infection. We have changed the final paragraph in the introduction to clarify this issue and the revised version of the manuscript appears as follows:

“(…)  The authors have developed a LBSap vaccine formulation, composed of L. braziliensis promastigote proteins plus saponin as an adjuvant, which proved harmless and safe for administration in murine, hamster, and canine models against L. infantum infection. Furthermore, this vaccine was able to induce an increase of total IgG, IgG1, IgG2 anti-Leishmania levels as well as CD5+, CD4+, and CD8+ T lymphocytes (TL) levels and CD21+ B lymphocytes in dogs [29-31]. Analyses of immunoprotection, using L. braziliensis as heterologous vaccine and L. infantum experimental challenge, demonstrated a lymphocyte activation profile (…)”.

  1. Line 203-204: typo? “12.000 glucosaminidase × g”

We are thankful for this observation and we have modified the sentence, as follows:

“(...) Briefly, after processing the skin, as described above, a part of the corresponding pellet was weighed, homogenized in 2 mL of buffer (0.1 M NaCl, 0.02 M Na3PO4, 0.015 M Na2-EDTA, pH 4.7), and centrifuged at 12.000 × g for 10 minutes at 4ºC (...)”.

  1. Lines 515-517: “In addition, the SMP2 group showed a significant increase in the CD3+CD4+ and CD3+CD8+ T-lymphocytes and B-lymphocytes”. Reference point should be clearly mentioned (SMP2-Ag vs naked SMP2).

We are thankful for this observation. We have corrected the sentence, as follows:

“(...) In addition, the SMP2Ag group showed a significant increase in the CD3+CD4+ and CD3+CD8+ T-lymphocytes and B-lymphocytes when compared to SMP2 group (…)”.

Reviewer 2 Report

I think some small discussion could be added about the choise of the skin for cytokine extraction 

Author Response

Comments and Suggestions for Authors:

I think some small discussion could be added about the choise of the skin for cytokine extraction.

We are thankful for this comment. We have added the requested information in Discussion section, as follows:

 “(…) Skin samples was collected in this study to analyze the inflammatory profile after inoculation with SMP particles, once they were administered intradermally. Moreover, the skin is the first point of contact between the vertebrate host and the parasite, in addition to be the first point of contact with the vector's immunomodulatory molecules. In this sense, it is extremely important to study the initial events triggered by vaccination in this compartment [7] (…). “

Reviewer 3 Report

This paper addresses the synthesis, characterization, and in vivo immunogenicity of chitosan-coated PDLLA sub-microparticles, and their application as adjuvant in an experimental vaccine against leishmaniasis.

Divided into three parts, find below my general concepts comments, and suggestions:

General concept comments

1.    The first part of the manuscript describes the synthesis of polysaccharide-coated SMP in high (5%) and low (1%) quantities and their biophysical characteristics.

·        In methods, item 2.2.1 (lines 115-133): resuspension protocol of the freeze-dried SMP is missing. Adequate lyophilization and resuspension protocols are critical to the colloidal stability of nano and microparticles. Please provide an explanation of these methods and supplementary data that certifies the particle’s stability before and after lyophilization.

·        Description of item “2.2.2. Determination of antigen encapsulation efficiency” is confusing and not clearly explained. Regarding the sub-micrometric size of the particles, would the SMP ultracentrifugation be sufficient to separate free antigen from SMP encapsulated/adsorbed one?

·        Line 276 - The terms “encapsulation efficiency” and “encapsulation percentage” can cause confusion to readers. It would be advised to use encapsulation percentage to describe results in the table. Please provide a clear explanation of payload meaning and calculation. Beware of overstating protein content using the BCA method to quantify samples contaminated with lipid and polymeric traces.

2.    The second part aims to check the inflammatory local effect of intradermal (?) injection of each formulation into mouse skin. Skin histology, cell infiltrate, MPO, NAG, and cytokines present in the local of injection were assessed.

·        There is an incoherence in the SMP route of administration: line 33 in the abstract says subcutaneous, line 167 in methods says intradermal and line 224 says subcutaneous. The different modes of adjuvant administration considerably change the cascade of events in the innate, local, immune response. Even though there are 2 different in vivo experiments with different readouts, please consider careful text correction, interpretation, and discussion to connect results from systemic immunogenicity (with SMP Ag) and in situ innate response (only SMP).

·        In item "2.3.1. Animals and immunization protocol", the experiment design lacks a positive adjuvant control. To prove the adjuvant effects, demonstrating selective cell recruitment at the application site, and to highlight the safety, tolerability, and non-toxic effects of the biodegradable SMPs, the authors should have considered using a commercially licensed adjuvant with a well-known mechanism of action (e.g. aluminium Al salts).

·        Line 172 – Describe the acronym CT.

·        Line 175 – For ethical reasons, consider reviewing this sentence. As the protocols for animal experimentation used have been approved for more than 10 years, please provide details of the anesthesia, sampling volumes, handling, and in vivo procedures to highlight that the principles of minimizing pain and suffering of the animals have been taken into account. 

·        Item "2.2.3.2. Histological examination", line 186: it would be great to have the numerical graphs of this counting represented by group to quantify and visualize statistical differences between skin cell infiltrate. This can be added in supplementary figures.

·        Line 291 – It is advised to summarize and focus the description on the main differences observed between groups to better understand.

·        In general, MPO, NAG and cytokines data are not consistent or conclusive – MPO production at day 2, and early IL-6 detection in the skin of Saline-injected mice confirm that there is an intrinsic inflammatory effect caused by the injection itself. It is advised to repeat assays to increase the n and confirm reproducibility.

3.    Finally, the authors performed an in vivo immunogenicity assay (murine model) comparing the injection of Ag-loaded SMP1 and SMP2 in the induction of circulating monocytes, NK, and lymphocytes (T and B).

·        Line 220: consider revision of repetitive “encapsulated” in item: "2.2.4. Systemic evaluation of encapsulated Leishmania braziliensis antigen encapsulated SMP"

·        Line 235 (paragraph) - "2.2.4.2. Immunophenotyping of blood cells by flow cytometry". Please provide the gate strategy in supplementary material and graphs with all points + medium to better visualize the cell profile distribution in each group.

·        Line 251- 15.000 events seem to be a very small sampling for such variable data. Consider acquiring more events next time. Use at least 50uL of blood per sample.

·        There is an inconsistency in the time point of the immunophenotyping analysis of peripheral blood immune cells by FACS: methods item 2.2.4.2. (line 236), results in 3.6 (line 377) and discussion (line 513). Please provide the right day for the readout of circulating immune cells after vaccination and consider discussion/conclusion revision based on the time point analyzed: innate and/or early immune events of adaptive response?

Discussion:

Line 417 – The surface charge of the SMP particles was not actually influenced by the concentration of chitosan/PDLLA in the final solution (ZP SMP1 = +71.6 ± 2.2 vs SMP2 = 68,4±1,7 is not statistical). Only the size is modified.

Line 424 – Showing ZP and size peaks, as well as their statistical comparisons for the different blank and loaded formulations (in 3 different experiments – supplementary material) would help to support that the chitosan is actually located on the polymeric micro-particle surface and that the antigen is adsorbed on it.

Conclusion:

On the whole, the use of SMPs with higher and lower concentrations was not clearly distinguished by a comparative analysis and a conclusion about the advantages of one or the other formulation for the vaccine application. It would be interesting to provide a clear view of each one should be better if a product development is proposed in the future.

The overall quality of language in this article rates 8/10, with satisfactory coherence and cohesion in English. 

Author Response

This paper addresses the synthesis, characterization, and in vivo immunogenicity of chitosan-coated PDLLA sub-microparticles, and their application as adjuvant in an experimental vaccine against leishmaniasis.

Divided into three parts, find below my general concepts comments, and suggestions:

General concept comments

  1. The first part of the manuscript describes the synthesis of polysaccharide-coated SMP in high (5%) and low (1%) quantities and their biophysical characteristics.

In methods, item 2.2.1 (lines 115-133): resuspension protocol of the freeze-dried SMP is missing. Adequate lyophilization and resuspension protocols are critical to the colloidal stability of nano and microparticles. Please provide an explanation of these methods and supplementary data that certifies the particle’s stability before and after lyophilization.

We add this protocol to the Methods section as follows:

“(…) After production, the NP formulations were aliquoted in small cryotubes stored under -80ºC and freeze-dried without cryoprotectants in freeze-drying equipment (Liobras, model 101, São Carlos, Brazil). In our previous unpublished work, we saw that chitosan acts as lyo- and cryoprotector of NP during freeze-drying). To test the formulations, all of them were resuspended in saline at an appropriate dilution to obtain 60 µg of antigen/100 µl (Antigen-loaded SMP), or to an equivalent mass of antigens (blank-SMP). They were sonicated to facilitate dispersion in an automatic ultrasonic water bath (Quimis, model Q335D2, Brazil) and used throughout the experiments in biological tests. (…)”.

Stability under storage was not performed because it has not been the main focus of our study. We plan to conduct this study during the formulation's scale-up in further development steps.

Description of item “2.2.2. Determination of antigen encapsulation efficiency” is confusing and not clearly explained.

We modify the text and equations to be clearer (please see the manuscript section 2.2.2)

 Regarding the sub-micrometric size of the particles, would the SMP ultracentrifugation be sufficient to separate free antigen from SMP encapsulated/adsorbed one?

Those SMP particles with sizes bigger than 350 nm are easily sedimented by ultracentrifugation at 20,000 × g. This is the most employed method to separate nanoparticles from soluble components in the water phase. Thus, antigen components non-associated with SMP remain in the supernatant.

Line 276 - The terms “encapsulation efficiency” and “encapsulation percentage” can cause confusion to readers. It would be advised to use encapsulation percentage to describe results in the table.

We agree and explain in the manuscript Method section and in Table 1 (table note). We only keep the “encapsulation efficiency” as it is a more honest way of expressing the Ag association with the SMP since it considers the loss of antigen during the SMP preparation process and gives us access to the real % of Ag associated with SMP (please see Section 2.2.2).

Please provide a clear explanation of payload meaning and calculation.

It is explained in the method section 2.2.2., equation 2. Payload is the ability of the particulate carrier to transport an active ingredient (wt/wt) %. In our case, it is the crude antigen.

Beware of overstating protein content using the BCA method to quantify samples contaminated with lipid and polymeric traces.

Following the method described in Section 2.2.2. to quantify protein in our samples, all polymers and lipids were removed by extraction with trichloromethane/water (1:1) and numerous steps of centrifugation and pooling of aqueous supernatant were undertaken. This procedure removes these SMP excipients from the aqueous phase where proteins were quantified. Furthermore, we have performed the same procedure with blank-SMP and no absorption was detected using the BCA method, as describes in the Section 2.2.2.

  1. The second part aims to check the inflammatory local effect of intradermal (?) injection of each formulation into mouse skin. Skin histology, cell infiltrate, MPO, NAG, and cytokines present in the local of injection were assessed.

There is an incoherence in the SMP route of administration: line 33 in the abstract says subcutaneous, line 167 in methods says intradermal and line 224 says subcutaneous. The different modes of adjuvant administration considerably change the cascade of events in the innate, local, immune response. Even though there are 2 different in vivo experiments with different readouts, please consider careful text correction, interpretation, and discussion to connect results from systemic immunogenicity (with SMP Ag) and in situ innate response (only SMP).

We are grateful for the careful reading of Reviewer #3. In fact, to assess the local inflammatory alterations we assayed in situ experiments using intradermal route for analysis of the compartmentalized immune response. In the second experiment, the subcutaneous compartment was used as route for vaccine administration since could represent more reliable way of immunization. We have made appropriate changes in the revised version of the manuscript to clarify this issue, as follows:

Section “2.2.3.1 Animals and immunization protocol”:

“(…) The intradermal route was used to provide a compartmentalized response and it was assessable for the in situ evaluation. (…)”.

Section “2.2.4.1 Immunization protocol”:

“(…) The subcutaneous route was used as a common compartment for immunization to trigger a systemic immune response.(…)”.

In item "2.3.1. Animals and immunization protocol", the experiment design lacks a positive adjuvant control. To prove the adjuvant effects, demonstrating selective cell recruitment at the application site, and to highlight the safety, tolerability, and non-toxic effects of the biodegradable SMPs, the authors should have considered using a commercially licensed adjuvant with a well-known mechanism of action (e.g. aluminium Al salts).

We are thankful for the Reviewer #2 comments. We agree with the Reviewer #2 but considering the ethics limitations about the number of mice used in the experiments we have included just the controls of news formulations without antigen (blank SMP1 and SMP2). Importantly, the SMPs evaluated were proposed as antigenic delivery system that need to work without the association of traditional adjuvants. However, we have described previously the action of L. braziliensis antigen associated or not with adjuvants as Saponin (Moreira et at. - Vet Immunol Immunopathol. 2009, 15;128(4):418-24. doi: 10.1016/j.vetimm.2008.11.030; Vitoriano-Souza et al. - PLoS One. 2012;7(7):e40745. doi: 10.1371/journal.pone.0040745). Moreover, we have analyzed the cellular recruitment and the innate immune response by combining saponin, monophosphoryl lipid-A and Incomplete Freund's Adjuvant with L. braziliensis antigen (Vitoriano-Souza et al. - Vaccine. 2019 Nov 20;37(49):7269-7279. doi: 10.1016/j.vaccine.2019.09.067.). Furthermore, the systemic effect of the immunization of L. braziliensis antigen in association or not with Saponin adjuvant was described in mice experiments that we have included additional commercial vaccines as “commercial control groups” (Leish-Tec® and Leishmune®) as reported previously (de Mendonça et al. - Parasit Vectors. 2016 30;9(1):472. doi: 10.1186/s13071-016-1752-6).

Line 172 – Describe the acronym CT.

The acronym CT was described, as follows:

“(…) To evaluate the effects caused by the response to the submicrometric particles, animals were divided into three experimental groups (n=5 animals/group/time): control (CT) group, inoculated with 100 µL/dose of saline; SMP1 group, inoculated with 100 µL/dose of SMP1formulation; and SMP2 group, inoculated with 100 µL/dose of SMP2 formulation. (…)”

Line 175 – For ethical reasons, consider reviewing this sentence. As the protocols for animal experimentation used have been approved for more than 10 years, please provide details of the anesthesia, sampling volumes, handling, and in vivo procedures to highlight that the principles of minimizing pain and suffering of the animals have been taken into account.

As requested, the sentence was modified. as shown below:

“(...) Animals were submitted to anesthesia using ketamine (200mg/Kg) and xylazine (16mg/Kg), for the removal of skin samples of the size of 0.5 cm. The animals were euthanized by cervical dislocation for histological analysis, cytokine and soluble proteins assessment. The clinical status of the animals was analyzed throughout the experiment, including behavioral aspects of the animals (pain or irritability) in addition to in situ changes. (...)”.

Item "2.2.3.2. Histological examination", line 186: it would be great to have the numerical graphs of this counting represented by group to quantify and visualize statistical differences between skin cell infiltrate. This can be added in supplementary figures.

We are thankful for this comment. In fact, the counting of cell infiltration could represent a more reliable form to identify the cell infiltration. We did these experiments a few years ago and lost the tissues to do the suggested analysis. However, in the first 72h of inoculum, the innate immune response is more prominent. For this reason, we used enzymatic biomarkers to assess neutrophil and macrophage infiltration as previously described.

Line 291 – It is advised to summarize and focus the description on the main differences observed between groups to better understand.

We revised the manuscript and adapted it according to your suggestions.

In general, MPO, NAG and cytokines data are not consistent or conclusive – MPO production at day 2, and early IL-6 detection in the skin of Saline-injected mice confirm that there is an intrinsic inflammatory effect caused by the injection itself. It is advised to repeat assays to increase the n and confirm reproducibility.

We are thankful for the Reviewer #2 comments. In fact, the inoculation of sterile saline can induce transient in situ changes. We have reported an infiltration by leucocytes and iNOS expression at first 96 h after saline administration in the inoculum site at the skin. In this sense, high levels of neutrophils (at 12h) and macrophages (at 48h and 96h) were described in mice (Vitoriano-Souza et al. - PLoS One. 2012;7(7):e40745. doi: 10.1371/journal.pone.0040745). Similar results were reported in hamsters (Moreira et at. - Vet Immunol Immunopathol. 2009, 15;128(4):418-24. doi: 10.1016/j.vetimm.2008.11.030). We believe that during the inoculation of saline, the stress on the extracellular matrix and cells could act as a mechanical stimulus that induces transient chances at the inoculum site including a slight inflammatory infiltrate.

We have included additional information in the Discussion section to clarify this issue as follows:

“(…) Despite the induction of IL-6 production and MPO in the inoculation site of control group it is possible to hypothesize that the mechanical stress on the extracellular matrix and cells can trigger slight transient inflammation, as previous reported [62, 63]. (…)”.

  1. Finally, the authors performed an in vivo immunogenicity assay (murine model) comparing the injection of Ag-loaded SMP1 and SMP2 in the induction of circulating monocytes, NK, and lymphocytes (T and B).

Line 220: consider revision of repetitive “encapsulated” in item: "2.2.4. Systemic evaluation of encapsulated Leishmania braziliensis antigen encapsulated SMP"

We have made appropriate changes in the revised version of the manuscript and the “encapsulated” has been replaced by “associated”.

Line 235 (paragraph) - "2.2.4.2. Immunophenotyping of blood cells by flow cytometry". Please provide the gate strategy in supplementary material and graphs with all points + medium to better visualize the cell profile distribution in each group.

We agree with Reviewer #3 and included the Supplementary Figure S1 in the Section “2.2.4.2 Immunophenotyping of blood cells by flow cytometry”.

Line 251- 15.000 events seem to be a very small sampling for such variable data. Consider acquiring more events next time. Use at least 50uL of blood per sample.

We thank the reviewer for the suggestion and we will modify the protocol for further tests.

There is an inconsistency in the time point of the immunophenotyping analysis of peripheral blood immune cells by FACS: methods item 2.2.4.2. (line 236), results in 3.6 (line 377) and discussion (line 513). Please provide the right day for the readout of circulating immune cells after vaccination and consider discussion/conclusion revision based on the time point analyzed: innate and/or early immune events of adaptive response?

We are thankful for the Reviewer #2 by detailed analysis of our manuscript. In fact, the first sentence of the Section “2.2.4.2 Immunophenotyping of blood cells by flow cytometry” was corrected as follows:

“(…) The blood cell immunophenotyping was performed by flow cytometry, 14 days after the third dose. (…)”.

Discussion:

Line 417 – The surface charge of the SMP particles was not actually influenced by the concentration of chitosan/PDLLA in the final solution (ZP SMP1 = +71.6 ± 2.2 vs SMP2 = 68,4±1,7 is not statistical). Only the size is modified.

First of all, we have to compare blank-SMP1 with Ag-loaded-SMP1 and zeta potential values are +71.6 ± 2.2 vs +57.0 ±1.5, respectively, and this difference is statistically significant (p < 0.001), as well as for SMP2, +68.4 ± 1.7 vs +58.1 ± 1.0, unloaded and Ag-loaded.

At fixed concentrations of chitosan in the SMP 1 (1%) and SMP 2 (5%) formulations (Table 1), the association of negatively charged antigens significantly alters the SMP zeta potential by approximately 10 mV in both formulations, probably by partially neutralize the positive charge of chitosan on the surface of the particles. As the polydispersion indices (PdI) of populations of SMP1 particles indicate broad size distributions, antigen association does not significantly change sizes. However, in blank SMP2 with narrow size distributions, sizes increase significantly with Ag association, as schematically represented in Figure 1.

The revised text was modified, as follows:

“(…) Chitosan-coated PLA SMPs prepared by the modified nanoprecipitation method has been successfully employed to provide two sizes of SMP loading crude antigen of L. braziliensis. Both were chosen for the in vivo studies. Ag-SMP1 and Ag-SMP2 with mean sizes of 416 nm and 938 nm and a similar surface charge, of approximately +57 mV. All the formulations showed positive zeta potential values, indicating that polycationic chitosan polymer is located at the SMP surface, as schematized in Figure 1. (…)”

Line 424 – Showing ZP and size peaks, as well as their statistical comparisons for the different blank and loaded formulations (in 3 different experiments – supplementary material) would help to support that the chitosan is actually located on the polymeric micro-particle surface and that the antigen is adsorbed on it.

We have included more detail and statistics in Table 1 and also a supplementary information with raw data of particle measurements.

Conclusion:

On the whole, the use of SMPs with higher and lower concentrations was not clearly distinguished by a comparative analysis and a conclusion about the advantages of one or the other formulation for the vaccine application. It would be interesting to provide a clear view of each one should be better if a product development is proposed in the future.

We are thankful for this comment. In fact, the results present in the study are preliminary, which demonstrate the promising use of polymeric particles in the field of vaccination against visceral leishmaniasis. However, further studies are needed to be able to suggest which formulation would be more promising for use in the development of new vaccines using an appropriate Leishmania challenge to identify the protection rate. In this sense, we have added the sentence, as follows:

Lines 576-578: “(...) However, in view of the results, it is not possible to highlight which of the two tested polymeric particles presents the best vaccine performance, since additional studies focusing on protection rate against L. infantum infection are required (…).”

Round 2

Reviewer 1 Report

I appreciate the authors' efforts to address all my comments.